# *Salmonella* Behavior in Meat during Cool Storage: A Systematic Review and Meta-Analysis

**DOI:** 10.3390/ani12212902

**Published:** 2022-10-23

**Authors:** Jorge Luiz da Silva, Bruno Serpa Vieira, Fernanda Tavares Carvalho, Ricardo César Tavares Carvalho, Eduardo Eustáquio de Souza Figueiredo

**Affiliations:** 1Federal Institute of Education, Science and Technology of Mato Grosso (IFMT), Cuiabá 78106-970, Brazil; 2Federal Institute of Education, Science and Technology of Mato Grosso (IFMT), Alta Floresta 78106-970, Brazil; 3Postgraduate Program in Animal Bioscience, University of Cuiabá, Cuiabá 78065-900, Brazil; 4Postgraduate Program in Animal Science, Federal University of Mato Grosso (UFMT), Cuiabá 78060-900, Brazil

**Keywords:** cold storage, growth rate, meat products, *Salmonella* spp.

## Abstract

**Simple Summary:**

*Salmonella* is an important pathogen associated with many foodborne disease outbreaks that can cause serious issues regarding public health, economic conditions, and quality of life, among others. Meat is the main human infection route for this bacterium, making food quality control in all production steps paramount. As *Salmonella* is a mesophilic bacterium, the cold chain is very important during meat processing, so pathogen behavior studies under cool storage, simulating the industry environment, can provide important data to the food industry. In this context, the aim of the present study was to perform a systematic review and meta-analysis of *Salmonella* behavior in meat during cool storage. Other conditions were also analyzed, such as meat sources (beef, chicken, pork, poultry, and turkey), fish, shellfish, media broth, package types, storage time, and bacterial inoculation (concentration and inoculation type).

**Abstract:**

The aim of the present study was to investigate *Salmonella* behavior in meat stored in cool conditions (between 0 °C and 7.5 °C), by employing a systematic review and meta-analysis. The data were obtained from research articles published in SciELO, PubMed, the Web of Science, and Scopus databases. The results of the retrieved studies were obtained from meat (beef, chicken, pork, poultry, and turkey), fish, shellfish, and broth media samples The data were extracted as sample size (*n*), initial concentration (*Xi*), final concentration (*Xf*), standard deviation (*SD*), standard error (*SE*), and microbial behavior effects (reduction or growth). A meta-analysis was carried out using the metaphor package from R software. A total of 654 articles were initially retrieved. After applying the exclusion criteria, 83 articles were selected for the systematic review, and 61 of these were used for the meta-analysis. Most studies were conducted at 0 °C to 4.4 °C storage temperatures under normal atmosphere package conditions. *Salmonella* Typhimurium, *S.* Enteritidis, and a cocktail (strain mixture) were inoculated at 5.0 and 6.0 log CFU mL^−1^. Articles both with and without the addition of antimicrobial compounds were found. *Salmonella* concentration decreases were observed in most studies, estimated for all study combinations as −0.8429 ± 0.0931 log CFU g^−1^ (95% CI; −1.0254, −0.6604) (*p* < 0.001), varying for each subgroup analysis. According to this survey, *Salmonella* concentration decreases are frequent during cool storage, although concentration increases and no bacterial inactivation were observed in some studies.

## 1. Introduction

*Salmonella* is an important pathogen, responsible for food disease outbreaks and termed salmonellosis. This microorganism can be found in the gastrointestinal tract of farm animals and can contaminate carcasses and meat when slaughter is carried out under inappropriate conditions.

This pathogen was responsible for 21.3% of the 2627 foodborne outbreaks that occurred between 2007 and 2019 in Brazil. It is important to highlight that 8998 outbreaks during this period were related, but the causative agents of only 29% were identified [1].

About 1.35 million illnesses, 26,500 hospitalizations, and 420 deaths occur each year in the United States due to foodborne outbreaks caused by *Salmonella* [2]. In Europe, a total of 91,662 and 94,425 salmonellosis cases were registered in 2016 and 2017, respectively, with *Salmonella* Enteritidis being the most common serotype reported [3].

Many food types can be contaminated by *Salmonella*, such as meat [4], fish and seafood [5,6], and fruit and vegetables [7,8]. *Salmonella* can also be present in processing environments (water, utensil and equipment surfaces, and handlers) [7]. One study indicated *Salmonella* contamination frequencies in Brazilian cattle carcasses and on their surfaces as 6.7% (6/90) and 2.6% (7/270), respectively [9].

It is essential to understand microorganism behavior in food to ensure food quality. Mathematical models have been developed in this regard to predict microbial risks in food products, describing microbial inactivation or growth according to intrinsic and extrinsic factors [10,11].

Storage temperature is a critical point of control in the food industry, as pathogens can grow at temperatures higher than 5 °C. One study [12] analyzed *S. enterica* behavior inoculated in poultry meat stored at 6 °C ± 2 °C for 35 days, reporting a 2.0 log CFU g^−1^ increase during the first seven days, followed by a 4.0 log CFU g^−1^ decrease at the end of the experiment. At 2 °C ± 2 °C, the concentration remained constant for two days, decreasing, thereafter, to 1.0 log CFU g^−1^, until undetectable from 25 to 35 days. Viable *Salmonella* concentrations have also been observed in frozen ground beef stored under a normal atmosphere for 5 to 75 days, following thawing in a refrigerator at 4 °C for 16 h [13].

Beyond the cold chain, the food industry has used natural compounds in the control of spoilage and pathogenic microorganisms. The addition of 0.5% cinnamon essential oil, for example, caused a 0.62 log CFU g^−1^ decrease in *Salmonella* Typhimurium concentrations in ground pork meat stored at 4 °C for 7 days [14], with higher antimicrobial concentrations causing more significant decreases in microorganism loads (*p* < 0.05). Olive extract also significantly affect *S.* Typhimurium. In another study, a 5.0 to 4.0 log CFU mL^−1^ decrease in *Salmonella* concentrations was observed in Muller Hilton broth containing 0.5% malic and acetic acid addition at 4 °C for 21 days [15]. In another assessment, the addition of 2.5% water–ethanol swamp cranberry and pomace extracts reduced pathogen concentrations in four log cycles in minced pork meat stored at 4 °C for 4 days under a normal atmosphere [16]. A decrease in *Salmonella* concentrations has also been observed in vacuum-packed ground beef stored at 3 °C for 12 days [17]. Decreases higher than 5 log CFU g^−1^ were observed for *S.* Typhimurium in vacuum-packed turkey meat treated with rosemary and oregano essential oils stored at 4 °C for 21 days [18]. Kahraman et al. [19], on the other hand, did not report decreases in *S.* Typhimurium concentrations in poultry meat containing 0.2% rosemary essential oil stored in modified-atmosphere packaging at 4 °C for 7 days.

These findings demonstrate discrepant results on *Salmonella* behavior in meat during refrigeration storage. Therefore, a systematic review and meta-analysis of *Salmonella* behavior in meat and its derivates under cooling storage and several environmental conditions, such as package type and antimicrobial compound addition, was carried out herein.

## 2. Material and Methods

The systematic review methodology has been registered on OSF (Open Science Framework) platform [20] (https://osf.io/8ayu2, accessed on 29 September 2022) under the doi registration https://doi.org/10.17605/OSF.IO/8AYU2.

Initially, three important observations were defined, namely population (sample), intervention or treatment, and measured outcome. Cold storage effects on *Salmonella* behavior (log CFU reduction or growth) in meat (beef, chicken, pork, poultry, and turkey), fish, shellfish, and in broth media were the specified populations. The measured outcome was derived from pathogen concentrations detected after cool storage.

### 2.1. Search Strategies

The research was performed in the SciELO, PubMed, Web of Science, and Scopus databases, using the following terms as a string: “(*Salmonella*) AND (Meat) AND ((growth) OR (survival) OR (Kinetic)) AND ((cold AND storage) OR (chill) OR (shelf AND life) OR (refrigerat*))”. No restriction filters were applied, and the terms were searched in retrieved paper titles, abstracts, and keywords.

The JabRef program (JabRef Team, US) [21] was used to organize the publications and identify duplicate articles. The selected articles included in this review employed storage temperatures between 0 °C and 7.5 °C.

### 2.2. Eligibility Criteria

Selected articles should be published in English or Spanish. Reviews, book chapters, and articles that did not use bacterial inoculation were excluded.

A second selection was performed by reading the titles and abstract, followed by full article reading. The third criterion was the use of an approved microbiological method for pathogen enumeration. *Salmonella* concentrations should be reported as log CFU per g, mL, or cm^2^. Experiments with results in MPN (most probable number) were excluded. As a fourth inclusion criterion for meta-analysis, the primary study must clearly describe the sample size and the standard deviations or errors for means. Several studies that did not report standard deviations or errors were used only for the systematic review. Experiments using ozone, high pressure, irradiation, and combined antimicrobial agents were excluded from the systematic review and meta-analysis. The selected articles were categorized as (1) no use of antimicrobial compounds and (2) use of antimicrobial compounds.

### 2.3. Data Extraction

Data of interest described in the articles were organized into spreadsheets by one reviewer. The extracted information included authors, year, sample type (meat or culture medium), package condition (normal atmosphere, vacuum, modified atmosphere), antimicrobial type, antimicrobial concentration, forms of antimicrobial application, *Salmonella* strain (subspecies or cocktail), inoculum concentration (in CFU g^−1^, CFU mL^−1^, CFU cm^−2^), inoculation type (surface and mixture), time (day) and temperature (°C) of storage, sample size (*n*), initial concentration (*Xi*), final concentration (*Xf*), standard deviation (*SD*), standard error (*SE*), and effect (reduction or growth) on microorganism behavior.

Gimp 2.10.8 (GIMP team) [22] and ImageJ (ImageJ team, US) [23] software were used to plot the data.

### 2.4. Statistical Analyses

Effect size was determined by the raw mean difference between the initial and final concentrations, as all the primary studies were reported on a log CFU scale. Each treatment was considered an individual observation.

Considering a primary study *j*, the effect size *θ* is the difference (RawDiff) in the sample means log reduction (*R*) or growth (*G*). Equation (1) was used for experiments without the addition of antimicrobial compounds, and Equations (2) and (3) were used for experiments with the addition of antimicrobial compounds.
(1)θ=RawDiff=Xf−Xi
(2)θ=RawDiff=Xcf−Xci
(3)θ=RawDiff=Xtf−Xti
where *Xi*, *Xci*, and *Xti* are the means of the initial concentration, initial concentration in the control samples, and initial concentration in the treated samples, respectively, while *Xf*, *Xcf*, and *Xtf* are the means of the final concentration, final concentration in the control samples, and final concentration in the treated samples, respectively.

The variance of mean log reduction or growth (VarRawDiff) was estimated as:(4)$$VarRawDiff=\sqrt{\frac{SDf^{2}}{nf}+\frac{SDi^{2}}{n}}$$
where *Sdf* and *Sdi* are the final and initial standard errors, respectively, *nf* and *ni* are the numbers of final and initial samples (repetitions), respectively.

When the article only provided the standard error (*SE*), this was transformed into *SD* through Equation (5).
(5)$$SD=SE.\sqrt{n}$$

The systematic review and meta-analyses results were separated into groups from 0 °C to 4.4 °C and 5.0 °C to 7.5 °C, as many scientific reports describe that *Salmonella* cannot survive at temperatures below 5 °C. The metafor R software package [24] was used to fit meta-analytic random-effects models.

## 3. Results

### 3.1. Systematic Review

A total of 654 articles published between 1985 and 2019 were selected following an electronic database search (Table 1).

A total of 155 duplicate articles were found. After initial title, abstract, and full article reading, 83 studies [12,13,14,15,16,17,18,19,24,25,26,27,28,29,30,31,32,33,34,35,36,37,38,39,40,41,42,43,44,45,46,47,48,49,50,51,52,53,54,55,56,57,58,59,60,61,62,63,64,65,66,67,68,69,70,71,72,73,74,75,76,77,78,79,80,81,82,83,84,85,86,87,88,89,90,91,92,93,94,95,96,97] were selected for the systematic review, and 61 of them were used in the meta-analysis (Figure 1), as the remaining articles did not present concentration means alongside standard errors or deviations. Several experiments employed normal, vacuum, and modified atmosphere package systems, also employing antimicrobial agents such as natural compounds, organic acids, seasoning, marination, and industrial sanitizers.

Data from 363 treatments were extracted from 83 selected articles in the systematic review. Most selected studies stored samples in a normal atmosphere between 0 °C and 4 °C for 11 to 35 days and applied a cocktail strain inoculation (Table 2). Articles with antimicrobial compound additions were also selected, and most cases used inoculum levels from 5.0 to 6.0 log CFU mL^−1^ of *Salmonella*.

### 3.2. Meta-Analyses Results

A total of 61 articles were selected for the meta-analysis, while 22 studies were excluded because they did not present means alongside standard deviations or standard errors.

*Salmonella* behavior was tested under many environmental conditions, such as different sample types (beef, chicken, pork, fish, and turkey), package conditions (normal, vacuum, and modified atmosphere packaging), inoculum concentrations, and binomial time versus temperature. Concentration decreases were observed in most studies, but a pathogen concentration increase or no changes were also noted.

The first meta-analysis with all the retrieved data (*n* = 61 articles/292 treatments) indicated high heterogeneity among studies (I^2^ = 97.52%, *p* < 0.0001), and the combined effect size was reduced by −0.8429 ± 0.0931 log CFU g^−1^ (95% CI: −1.0254, −0.6604). The combined meta-analysis result (*k* = 292 treatments) presented a significant effect (*p* < 0.001), demonstrating that cool temperatures can control and decrease *Salmonella* concentrations.

Decreased *Salmonella* concentrations were observed in most treatments, independent of package condition and antimicrobial compound addition (Table 2). Decreased between 0.1 and 2 log CFU g^−1^ or mL^−1^ were more frequently reported, although some increases of less than 1 log CFU g^−1^ were also observed.

A funnel plot graphic was prepared to verify potential publication bias between the results (Figure 2).

The first subgroup effect analysis was observed as a function of sample, categorized as beef, chicken, pork, turkey, and fish. Beef analyses were performed in 107 treatments reported in 23 studies (Figure 3), while chicken and pork samples were used in 60 (17 articles) and 37 (10 articles) treatments, respectively (Figure 4 and Figure 5). Turkey and fish samples were employed in 25 (2 articles) and 20 (two articles) treatments, respectively.

The package effect was the follow subgroup, and the retrieved studies were categorized as storage under normal atmosphere packages (NA: 211 treatments from 43 articles), vacuum packages (VC: 34 treatments from 16 articles), and modified atmosphere packages (MAP: 27 treatments from 8 articles) (Figure 6 and Figure 7, respectively).

Another subgroup consisted of effects analyses as a function of two storage temperature intervals, from 0 °C to 4.4 °C and 5 °C to 7.5 °C. These temperature intervals were selected because temperatures lower than 5 °C are a challenge for microorganisms, and a maximum temperature at 7 °C is recommended for refrigerated storage. Data from 226 treatments extracted from 48 studies were analyzed for the group stored at 0 °C to 4.4 °C, while data from 66 treatments extracted from 17 articles were analyzed for the group stored at 5 °C to 7.5 °C (Figure 8).

Subgroup analyses were also performed on *Salmonella* concentration results as a function of the already referenced storage temperature intervals (Table 3).

## 4. Discussion

### 4.1. Systematic Review

Storage temperatures between 0 °C and 4 °C (77.97%) were more commonly applied compared to temperatures between 5 °C and 7.5 °C (22.03%), probably because the retrieved studies aimed to observe *Salmonella* behavior at low temperatures.

In general, most treatments comprised beef samples (23.96%), antimicrobial compound addition (38.84%), normal packaging (53.99%), more than 10 storage days (47.10%), 6 log CFU mL^−1^ inoculum concentration (25.61%), and *Salmonella* Typhimurium strains (32.23%) (Table 2).

Beef (24.4%), chicken (18.18%), and pork (12.95%) were the major samples used at 0 °C to 4 °C storage temperatures. Under these conditions, studies were performed using antimicrobial compounds (38.84%), normal atmosphere (53.99%), and storage times of less than 10 days (47.10%).

In studies conducted at 0 °C to 4 °C, *Salmonella* Typhimurium was the most inoculated strain (32.23%), and the most frequent inoculum concentrations were 5, 6, and more than 6 log CFU mL^−1^.

At 5 °C to 7.5 °C storage temperatures, beef was the most common sample (13.49%), with antimicrobial compound addition (9.36%) and normal atmosphere packing (16.52%), during 11 to 35 storage days using cocktail and *S.* Typhimurium strains (7.43%) at a 3 log CFU mL^−1^ inoculum concentration (7.71%).

As a result, decreases lower than 1 log CFU g^−1^ in the control (18.73%) and treated (11.29%) groups at 0 °C to 4 °C storage were the most frequent. A reduction of more than 2 log CFU g^−1^ was also observed at both temperatures. However, about 19.28% of the analyzed treatments presented growth rates of 5.23% and 3.58% at 0 °C to 4 °C, respectively.

At temperature intervals from 0 °C to 4 °C, the control group treatments presented 18.73% decreases less than 1 log CFU g^−1^ and 21.48% decreases between 1 log CFU g^-1^ to 2 log CFU g^−1^ in pathogen level. From 5 °C to 7.5 °C, the main results were 5.78% reduction <1 log CFU g^−1^ and 3.03% reductions both >1 log CFU g^−1^ and >2 log CFU g^−1^.

Most articles analyzed bacterial behavior under a normal atmosphere and demonstrated pathogen reduction. Experiments with vacuum packing and MAP also resulted in *Salmonella* decreases in most cases, although growth-rate values were noted in all conditions.

Although most selected studies reported decreases in *Salmonella* concentrations, some indicated increased pathogen concentrations even under cool storage. One study, for example, observed a reduction between 1.4 and 1.9 log CFU g^−1^ in *Salmonella* concentrations in ground beef packed under a vacuum and in a modified atmosphere at 3 °C for 12 days [17]. In another study, 0.17 and 0.97 log CFU *Salmonella* -concentration decreases were detected in modified-atmosphere packed (MAP) beef stored at 7.5 °C for 12 days, although a 1.69 log CFU g^−1^ pathogen reduction was observed in vacuum-packed beef in the same study [42].

A decrease in *Salmonella* concentrations was observed in chicken samples containing 2% acetic acid at 2 °C, 6 °C, and 8 °C for 9 days, with reduction values of 0.7, 0.9, and 0.9 log CFU/g, respectively [61]. However, in another study, *Salmonella* Typhimurium growth was observed in vacuum-packed minced chicken treated with olive oil and stored at 2 °C for 60 days [43].

Increasing *Salmonella* concentrations in vacuum-packed whole shrimp with potassium sorbate, sodium benzoate, sodium diacetate addition and control were observed at 4 °C for 7 days, ranging from 0.93 to 1.84 log CFU g^−1^ [96]. Edwards et al. [40] also reported decreases between 0.01 and 0.23 log CFU g^−1^ in shrimp inoculated with *S.* Typhimurium, *S.* Enteritidis, and *S.* Infantis stored at 4 °C for 2 days. The authors, however, also observed growth of approximately 0.10 and 0.13 of the same *Salmonella* strains in the same storage conditions.

Regarding challenge tests with antimicrobial compounds, *Salmonella* concentration decreases were observed with malic and acetic acid addition in Mueller Hilton broth stored at 4 °C for 21 days [16]. In another study, decreases of 1.6 and 0.37 log CFU g^−1^ in *Salmonella* Typhimurium concentrations were detected in BHI broth stored at 4 °C and 7 °C, respectively, for 7 days [69]. Silva et al. [88] reported a pathogen decrease of over 6.0 log CFU g^−1^ using chitosan coating in inoculated beef stored at 4 °C for 3 days. *Salmonella* concentrations were also reduced by approximately 1.8 and 1.6 log cycles following the addition of 0.3% carvacrol and thymol essential oils in beef stored at 4 °C for 7 days [68].

A study comprising lactic acid or acidified sodium chlorite addition in minced beef stored at 5 °C for 14 days reported a 0.031 to 0.264 log-cycle reduction in *Salmonella* cocktail concentrations, although an increase was observed in the control treatment of 0.8 log CFU g^−1^ bacterium concentrations [50]. A *Salmonella* Typhimurium concentration reduction higher than 3 log CFU g^−1^ was reported for minced meat stored at 4 °C for 7 days [67]. Nisiotou et al. [74] also indicated a decrease in *Salmonella* Typhimurium concentrations in modified-atmosphere-packaged beef, with or without marination, stored at 5 °C for 19 days of 0.6 and 2.90 log CFU g^−1^ in control and treated samples, respectively.

### 4.2. Meta-Analysis

The funnel plot graph (Figure 2) presents the dispersion noted for the selected data, with the most observed outcomes varying between approximately −3 and 1 log CFU g^−1^, demonstrating result variability and the absence of publication bias.

The beef sample effect was −0.9951 ± 0.1524 log CFU g^-1^ (95% CI: −1.2937, −0.6964), and a reduction of −0.8943 ± 0.1880 log CFU/g (95% CI: −1.2628, −0.5258) was found for the pork samples. However, studies on chickens indicated a reduction of −0.1639 ± 0.1994 log CFU g^−1^ (95% CI: −0.5548, 0.2270), although the pathogen growth rate was observed within a sample confidence interval (*p* = 0.4111) according to the combined meta-analysis result.

For turkey and fish samples, reductions of −1.0140 ± 0.1495 log CFU/g (95% CI: −1.3070, −0.7211) and −1.8090 ± 0.1924 log CFU/g (95% CI: −2.1860, −1.4319) were observed, respectively. Significant effects (*p* < 0.001) in the meta-analyses results were presented in experiments using beef, pork, turkey, and fish.

The second subgroup comprised antimicrobial compound addition, categorized into groups with (152 treatments/37 articles) and without (142 treatments/55 articles) antimicrobial compound addition compounds or controls. Both studies demonstrated a significant effect (*p* < 0.001), but antimicrobial addition was reduced by −1.2041 ± 0.1559 log CFU g^−1^ (95% CI: −1.5096, −0.8986), while in the control group or in treatments without antimicrobial compounds, a −0.4526 ± log CFU g^−1^ (95% CI: −0.6282, −0.2769) was observed. This demonstrated the importance of studying antimicrobial compound effects, mainly those of natural compounds, against *Salmonella* in meat.

The normal atmosphere and modified atmosphere packaging displayed more significant *Salmonella* concentration reductions, of −0.8446 ± 0.1056 log CFU g^−1^ (95% CI: −1.0515, −0.6376) (*p* < 0.001) and −0.8604 ± 0.3273 log CFU g^−1^ (95% CI: −1.5018, −0.2190) (*p* < 0.01). No significant effect was observed for vacuum-packaged samples (*p* = 0.3934), with a combined value of −0.2506 ± 0.2936 log CFU g^−1^ (95% CI: −0.8260, 0.3249).

A decrease in *Salmonella* concentrations was observed in both temperature intervals, albeit with a difference in *p* values and higher pathogen concentration decreases at lower temperatures.

The general effect at 0 °C to 4.4 °C temperature storage was a −0.9217 ± 0.1063 log CFU g^−1^ (95% CI of −1.1301, −0.7132) (*p* value < 0.001) decrease, while *Salmonella* concentrations at 5 °C to 7.5 °C were reduced by −0.5742 ± 0.1955 log CFU g^−1^ (95% CI of −0.9574, −0.1910) (*p* value < 0.01). Temperatures from 0 °C to 4 °C promoted more significant pathogen concentration decreases, with a statistically significant effect.

Concerning all subgroups, a higher *Salmonella* concentration reduction was observed for fish, under both normal and MAP atmospheres, with antimicrobial compound addition and stored between 0 °C and 4.4 °C.

As expected, decreases were noted in all analyses except for treatments using broth samples and MAP packages at 5 °C to 7.5 °C (interval 2), with treatments analyzed at 0 °C to 4 °C (interval 1) presenting more significant *Salmonella* concentration decreases. At 0 °C to 4 °C, only vacuum-packaged pork samples with up to 4 log CFU mL^−1^ pathogen concentration treatments presented no significant statistical effect, and, at 5 °C to 7.5 °C, these effects were observed in less than half of the studies.

The more significant effects in both temperature intervals were observed for studies employing fish samples, with −1.93 and 1.34 log CFU g^−1^ decreases at intervals 1 and 2, respectively. When considering studies with K > 60, beef samples and antimicrobial compound addition treatments exhibited more significant effects, with −1.24 and −1.25 log CFU g^−1^ values, respectively.

Thus, differential statistical effects were observed between treatments according to storage temperature intervals. For example, beef samples presented a −1.2 log CFU g^−1^ (*p* < 0.001) decrease at interval 1 and a −0.59 log CFU g^−1^ (*p* < 0.05) decrease at interval 2. In addition, this same condition was observed in many other results. Only studies on fish presented the same statistical effect at both temperature intervals (*p* < 0.001). It is important to note that chicken samples did not present significant effects at either temperature interval.

Experiments without antimicrobial compound addition and under vacuum packaging and MAP packaging are less effective in controlling *Salmonella* growth at interval 2. Decreases in *Salmonella* concentrations were lower for treatments using the *S.* Typhimurium and *S.* Enteritidis mixture inoculation type, up to 4 log CFU g^−1^ for 10 days, compared to interval 1 treatments.

## 5. Conclusions

*Salmonella* concentration decreases were observed in meat under cool storage, which were higher at lower storage temperatures. According to this meta-analysis, other factors also contribute to *Salmonella* concentration decreases during refrigerated storage time, such as beef and fish samples, normal atmosphere and MAP, antimicrobial compound addition, and storage from 0 °C to 4.4 °C.

Cool storage was effective for *Salmonella* growth control, where decreases in meat between −0.15 and −1.24 log CFU g^−1^ were observed at colder temperatures, demonstrating the importance of cold chains for both the industry and customers.

Although most experiments reported *Salmonella* decreases during cool storage, the pathogen was not eliminated in the samples. Thus, the risk of salmonellosis transmitted by meat remains and should be considered.

## Figures and Tables

**Figure 1 animals-12-02902-f001:**
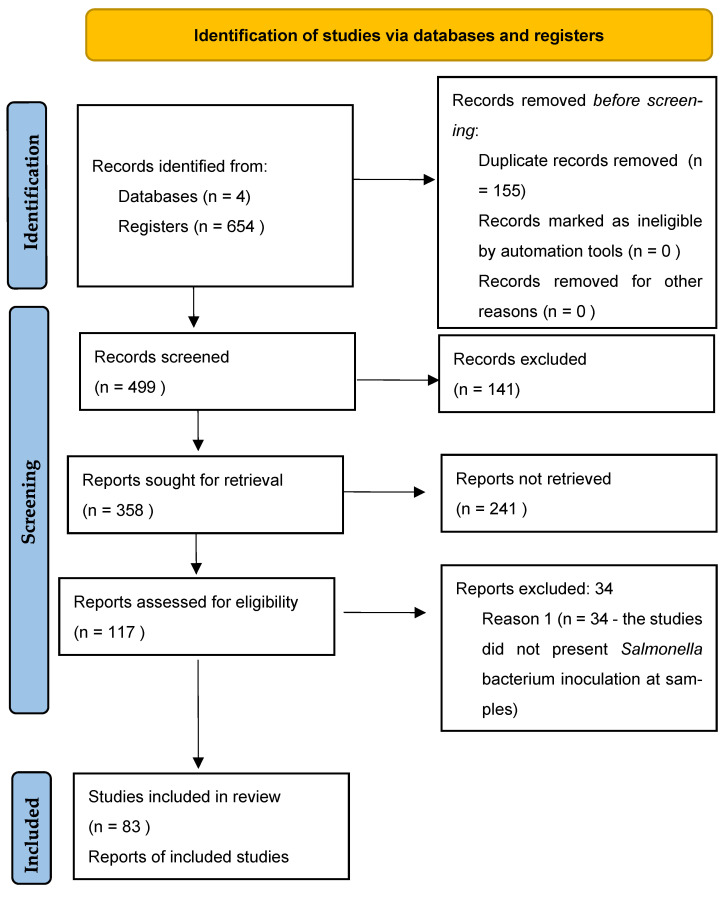
Search diagram and article selection criteria for the systematic review and meta-analysis according to PRISMA [98].

**Figure 2 animals-12-02902-f002:**
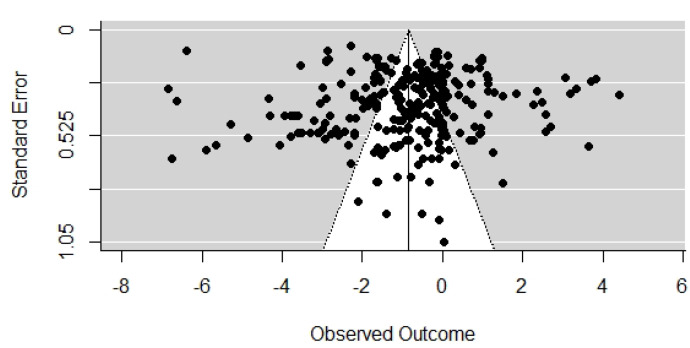
Funnel plot graphic concerning *Salmonella* behavior data in meat and media under cool storage.

**Figure 3 animals-12-02902-f003:**
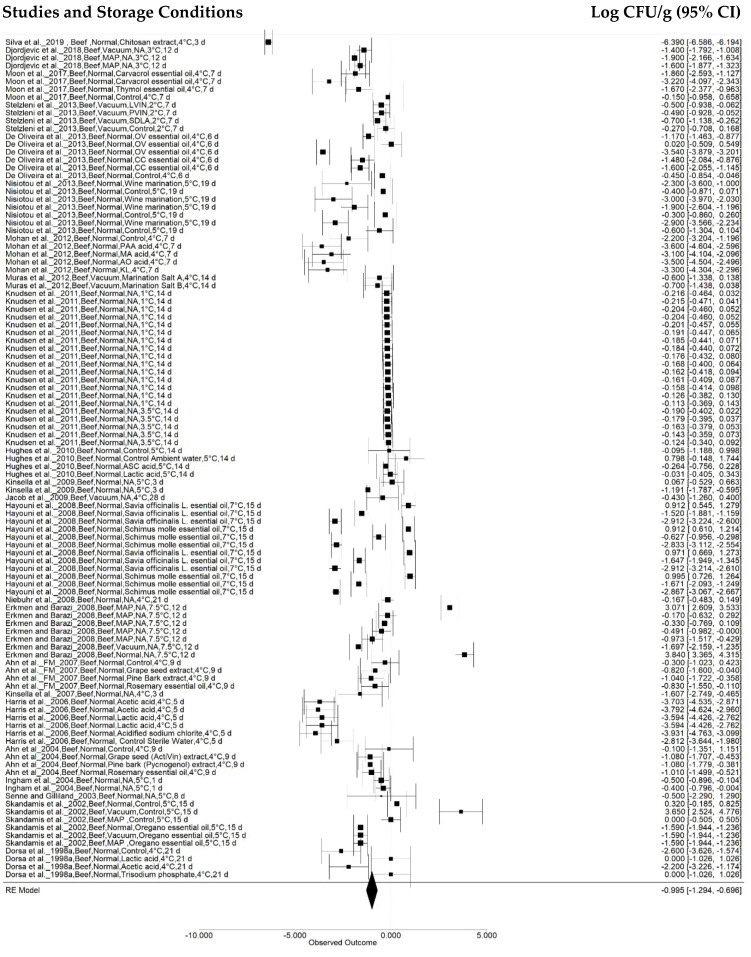
Forest plot of data on *Salmonella* concentration effects in beef stored at 0 °C to 7.5 °C (*n* = 23 articles/107 treatments) (I^2^ = 98.48%, *p* < 0.0001).

**Figure 4 animals-12-02902-f004:**
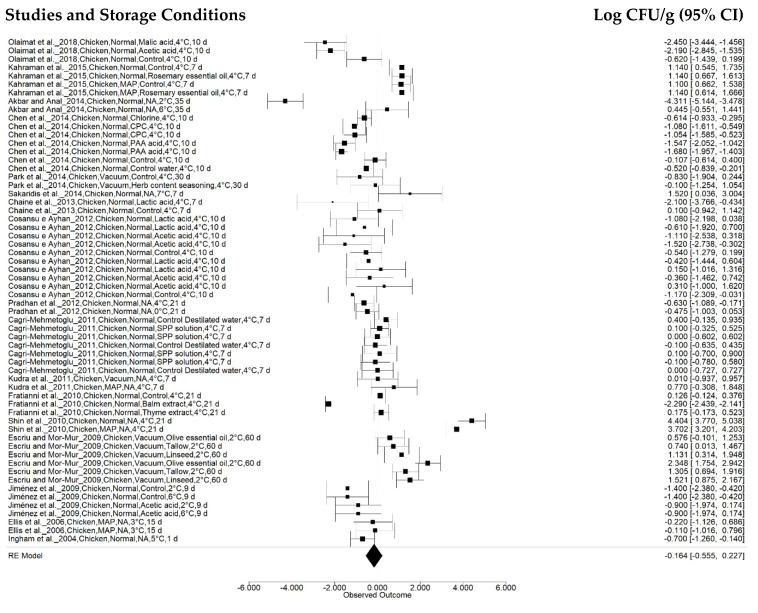
Forest plot concerning *Salmonella* concentration effects in chicken stored at 0 °C to 7.5 °C (*n* = 17 articles/60 treatments) (I^2^ = 96.60%, *p* = 0.4111).

**Figure 5 animals-12-02902-f005:**
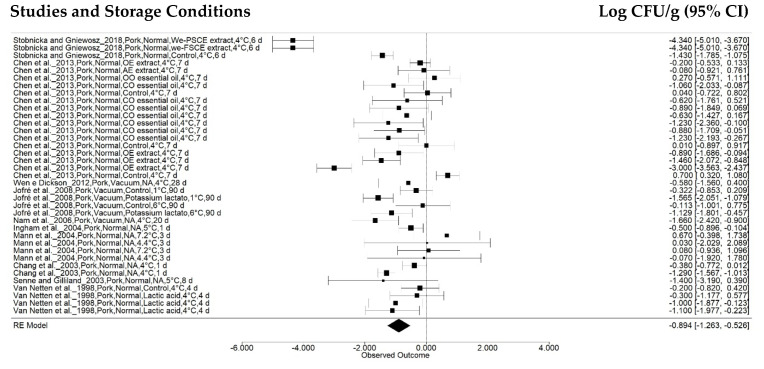
Forest plot concerning *Salmonella* concentration effects in pork stored at 0 °C to 7.5 °C (*n* = 10 articles/37 treatments) (I^2^ = 91.70%, *p* < 0.0001).

**Figure 6 animals-12-02902-f006:**
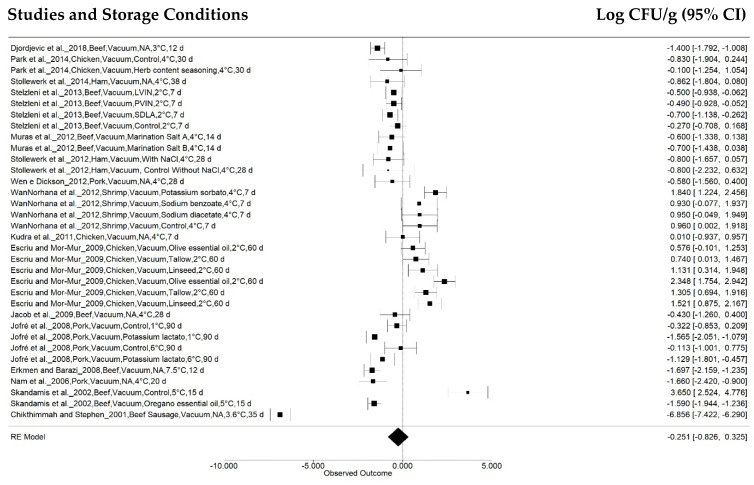
Forest concerning *Salmonella* concentration effects in meat stored at 0 °C to 7.5 °C in vacuum packages (*n* = 16 articles/34 treatments) (I^2^ = 96.45%, *p* = 0.3934).

**Figure 7 animals-12-02902-f007:**
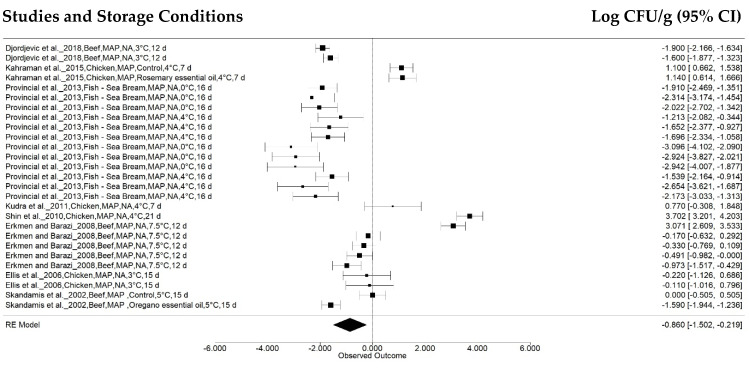
Forest plot concerning *Salmonella* concentration effects in meat stored at 0 °C to 7.5 °C in modified atmosphere packages (*n* = 8 articles/27 treatments) (I^2^ = 97.38%, *p* < 0.01).

**Figure 8 animals-12-02902-f008:**
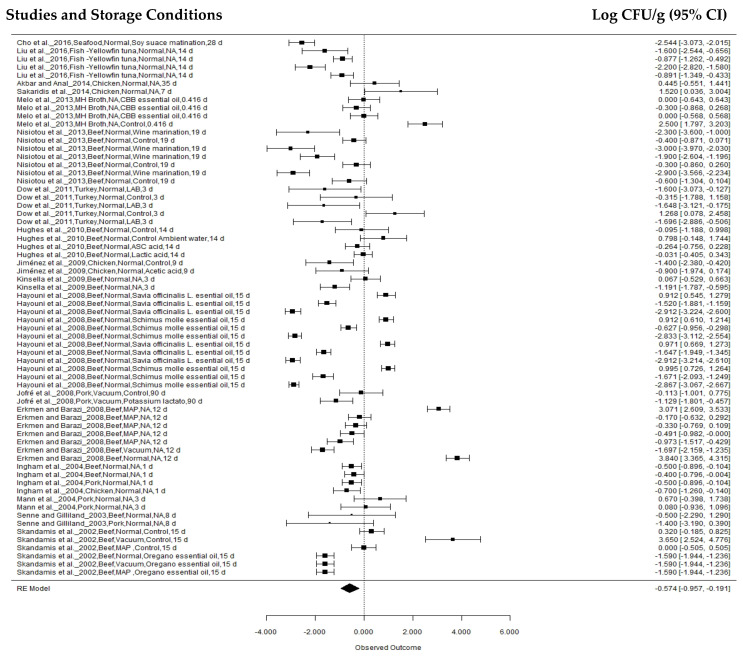
Forest plot concerning *Salmonella* concentration effects in meat under cool storage at 5 °C to 7.5 °C (*n* = 17 articles/226 treatments) (I^2^ = 97.59%, *p* < 0.001).

**Table 1 animals-12-02902-t001:** Number of articles retrieved through the database search.

Database	Number of Articles
SciELO	117
PubMed	184
Web of Science	139
Scopus	214
Total	654

**Table 2 animals-12-02902-t002:** Data on *Salmonella* behavior during cool storage were extracted from selected articles for the systematic review.

Extracted Data (*n* = 83 Studies)		Temperatures		Total Treatments
		0 °C to 4.4 °C (k = 283 Treatments)	5 °C to 7.5 °C (k = 80 Treatments)	
Sample	Beef	87 (23.96%)	50 (13.77%)	363
	Chicken	66 (18.18%)	6 (1.65%)	
	Pork	47 (12.94%)	6 (1.65%)	
	Turkey	24 (6.61%)	5 (1.37%)	
	Fish	16 (4.40%)	4 (1.10%)	
	Seafood	19 (5.23%)	1 (0.27%)	
	Broth	9 (2.47%)	6 (1.65%)	
	Other	15 (4.13%)	2 (0.55%)	
Treatments	Treatments *	95 (26.17%)	30 (8.26%)	363
	Control **	47 (12.94%)	16 (4.40%)	
	Antimicrobial **	141 (38.84%) (72 S)	34 (9.36%) (21 S)	
Packing	Normal	196 (53.99%)	60 (16.52%)	363
	Vacuum	51 (14.04%)	8 (2.20%)	
	MAP	22 (6.06%)	7 (1.93%)	
	Media (broth)	14 (3.85%)	5 (1.37%)	
Storage time (days)	≤10 d	171 (47.10%)	27 (7.43%)	363
	11 ≤ d ≥ 35 d	94 (25.89%)	48 (13.22%)	
	>35 d	18 (4.95%)	5 (1.37%)	
Serovar	Cocktail	65 (17.90%)	27 (7.43%)	363
	Enteritidis	52 (14.32%)	13 (3.58%)	
	Typhimurium	117 (32.23%)	27 (7.43%)	
	Others	49 (13.49%)	13 (3.58%)	
Inoculation level (log CFU mL^−1^)	1.0 and 2.0	2 (0.55%)	7 (1.92%)	363
	3.0	15 (4.13%)	28 (7.71%)	
	4.0	44 (12.12%)	13 (3.58%)	
	5.0	66 (18.18%)	15 (4.13%)	
	6.0	93 (25.61%)	14 (3.85%)	
	>6.0	63 (17.35%)	3 (0.82%)	
Storage effect on control treatment	Growth (>2 log)	3 (0.82%)	4 (1.10%)	363
	Growth (>1 log)	3 (0.82%)	2 (0.55%)	
	Growth (<1 log)	19 (5.23%)	8 (2.20%)	
	Reduction (<1 log)	68 (18.73%)	21 (5.78%)	
	Reduction (>1 log)	25 (6.88%)	6 (1.65%)	
	Reduction (>2 log)	22 (6.06%)	3 (0.82%)	
	Not changed	2 (0.55%)	2 (0.55%)	
Storage effect on antimicrobial treatment	Growth (>2 log)	7 (1.92%)	0 (0.00%)	
	Growth (>1 log)	6 (1.65%)	0 (0.00%)	
	Growth (<1 log)	13 (3.58%)	5 (1.37%)	
	Reduction (<1 log)	41 (11.29%)	5 (1.37%)	
	Reduction (>1 log)	32 (8.81%)	11 (3.03%)	
	Reduction (>2 log)	37 (10.19%)	11 (3.03%)	
	Not changed	5 (1.37%)	2 (0.55%)	

* experiments without antimicrobial compound addition; ** experiments with antimicrobial compound addition: control and antimicrobial treatments (N = natural antimicrobial, S = synthetic antimicrobial).

**Table 3 animals-12-02902-t003:** Subgroup analyses concerning *Salmonella* concentration data as a function of temperature intervals (0 °C to 4.4 °C/5 °C to 7.5 °C).

Subgroup		Storage Temperatures					
		0 °C to 4.4 °C			5 °C to 7.5 °C		
		Effects (log CFU/g ± Se) (95% CI)	I^2^ (%)	*p*-Value	Effects (log CFU/g ± Se) (95% CI)	I^2^ (%)	*p*-Value
Sample	Beef (K = 66 and 41)	−1.2423 ± 0.1908 [−1.6163, −0.8684]	98.56	***	−0.5966 ± 0.2600 [−1.1063, −0.0869]	98.39	*
	Chicken (K = 55 and 5)	−0.1567 ± 0.2108 [−0.5700, 0.2565]	96.87	0.4572	−0.3200 ± 0.4213 [−1.1458, 0.5059]	73.04	0.4476
	Pork (K = 31 and 6)	−0.9883 ± 0.2140 [−1.4077, −0.5690]	92.74	***	−0.3873 ± 0.2565 [−0.8900, 0.1154]	54.04	0.1310
	Fish (K= 16 and 4)	−1.9339 ± 0.2286 [−2.3818, −1.4859]	83.62	***	−1.3413 ± 0.3165 [−1.9615, −0.7211]	80.07	***
	Turkey (K= 20 and 5)	−1.0523 ± 0.1536 [−1.3534, −0.7512]	80.23	***	−0.7743 ± 0.6247 [−1.9988, 0.4501]	75.72	0.2152
	Broth (K = 3 and 4)	−3.5328 ± 1.7255 [−6.9146, −0.1509]	97.93	*	0.5373 ± 0.5992 [−0.6371, 1.7116]	93.10	0.3699
Package	Normal (K =161 and 50)	−0.8666 ± 0.1196 [−1.1010, −0.6322]	97.62	***	−0.7720 ± 0.2280 [−1.2188, −0.3252]	97.74	***
	Vacuum (K = 29 and 5)	−0.2512 ± 0.3315 [−0.9009, 0.3985]	96.56	0.4485	−0.2741 ± 0.6339 [−1.5165, 0.9683]	95.35	0.6654
	MAP (K = 22 and 5)	−1.1129 ± 0.3519 [−1.8025, −0.4232]	97.00	**	0.2235 ± 0.7322 [−1.2116, 1.6585]	97.79	0.7602
Antimicrobial	With (k = 122 and 30)	−1.1911 ± 0.1912 [−1.5658, −0.8164]	97.96	***	−1.2572 ± 0.2807 [−1.8076, −0.7074]	98.20	***
	Without (K = 71 and 24)	−0.7376 ± 0.1199 [−0.9726, −0.5026]	96.07	***	−0.1628 ± 0.3071 [−0.7647, 0.4390]	96.32	0.5959
Serovar	Cocktail (K = 58 and 19)	−1.0364 ± 0.1385 [−1.3079, −0.7649]	92.62	***	−0.4163 ± 0.1982 [−0.8046, −0.0279]	83.14	*
	Typhim. (K = 91 and 24)	−0.7557 ± 0.1637 [−1.0766, −0.4348]	96.46	***	−0.4878 ± 0.3458 [−1.1654, 0.1899]	97.34	0.1584
	Enteriti. (K = 48 and 10)	−1.1029 ± 0.1923 [−1.4798, −0.7260]	95.38	***	−0.5041 ± 0.5966 [−1.6733, 0.6652]	99.10	0.3981
Inoculation type	Mixture (K = 69 and 27)	−1.2321 ± 0.1569 [−1.5396, −0.9247]	95.34	***	−0.4277 ± 0.3105 [−1.0363, 0.1809]	98.55	0.1684
	Surfa. (K = 157 and 39)	−0.7828 ± 0.1344 [−1.0462, −0.5195]	97.91	***	−0.6779 ± 0.2394 [−1.1471, −0.2087]	95.39	**
Level	Up 4 log (K = 51 and 33)	−0.2101 ± 0.1335 [−0.4717, 0.0515]	96.55	0.1154	−0.4656 ± 0.3017 [−1.0570, 0.1257]	98.65	0.1227
	> 4 log (K = 175 and 33)	−1.1305 ± 0.1446 [−1.4140, −0.8470]	97.42	***	−0.680 ± 0.1874 [−1.0494, −0.3147]	89.58	***
Storage time	Up 10 d (K = 150 and 22)	−0.9794 ± 0.1612 [−1.2953, −0.6636]	97.51	***	−0.2729 ± 0.1907 [−0.6466, 0.1008]	81.22	0.1523
	>10 d (K = 76 and 44)	−0.8010 ± 0.1350 [−1.0657, −0.5364]	97.21	***	−0.7017 ± 0.2504 [−1.1924, −0.2110]	98.30	**

*** *p* < 0.001,** *p* < 0.01,* *p* < 0.05, Se = standard error.

## Data Availability

Not applicable.

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
