# Peer review of "Salmonella Behavior in Meat during Cool Storage: A Systematic Review and Meta-Analysis"

_animals, 2022, doi:10.3390/ani12212902_

Round 1
Reviewer 1 Report
The manuscript is very well prepared and even if the results of the analyzes do not bring any fundamental knowledge, their complexity and clear presentation should be appreciated.
I recommend publishing the manuscript after correcting minor flaws:
· on line 21 put the term Salmonella in italics
· on lines 28-29, I recommend dividing: meat (beef, chicken, pork, poultry, turkeys), fish and shellfish and broth media samples. The same for lines 101-102.
· On line 39: CFU instead of UFC. The same for lines 236, 375, 380
· on lines 47 and 51, I recommend using the term foodborne outbreaks
· on line 130, I recommend using the term non-preservative instead of the term normal for the packaging conditions, or normal atmosphere
· line 250: …turkey instead of Turkey
· on line 290, I recommend changing it to: ...and maximum temperature at 7 °C is recommended for refrigerated storage.
· up to chapter 4, temperatures of 0-4.4 °C are given, but in chapter 4 Discussion, 4 °C is given. I recommend unifying, it is best to indicate the range 0-4 °C, because we mathematically round 4.4 to 4.
· Line 344: A decrease instead of Decrease
· On line 369 author Nisiotou, but in References on line 638 Nisiotu. What is correct?
· Finally, one more term for discussion: the authors use the term concentration in relation to salmonella. But this word is used to express the content of chemical substances. Isn't it better to use the term level or levels, as shown on lines 220-221?
Author Response
Dear reviewer,
Thank you very much for your contribution and very relevant suggestions in this manuscript. The considerations were accepted and certainly raised the scientific level of this article. The modifications are highlighted in yellow stripes on the text.
Please see the attachment.

Reviewer 2 Report
The manuscript presented by the authors is a very interesting review about the effect of cooling storage and other parameters on the Salmonella behaviour in meat and its derivates. The paper has an overall high quality and scientific soundness. The only suggestion I have concerns the english language: some sentences are not clear and there are some typos and incorrect forms in the manuscript such as:
- L18: chain not chair
- L32: at first not In the first time
- L95: derivates not derivatives
and many others. I suggest to have the all manuscript revised by an english mother tongue.
Author Response
Dear reviewer,
Thank you very much for your contribution and very relevant suggestions in this manuscript. The considerations were accepted and certainly raised the scientific level of this article. The modifications are highlighted in yellow stripes on the text.
Please see the attachment

Reviewer 3 Report
The paper “Salmonella behavior in meat during cooling storage: a 2 systematic review and meta-analysis” is an interesting review about Salmonella behavior after experimental inoculation in meat and fish under different refrigeration temperature and time storage, packaging conditions, with or without the use of antimicrobials. The review is sound and well written. Selection criteria for articles are valid and accurate. Results are well presented. I have just one comment, which is more a curiosity. Current legislation in European Union (Reg. CE n. 853/2004 https://eur-lex.europa.eu/legal-content/EN/TXT/PDF/?uri=CELEX:32004R0853&qid=1664788743750&from=IT) distinguish between “meat” and “fishery products” thus making not possible to use a unique definition. In this review the authors speak generically about “meat” including also fish. I wonder if such distinction is present in Brazilian legislation. In this case, the authors should specify that the review regards meat AND fish.
Author Response
Dear reviewer,
Thank you very much for your contribution and very relevant suggestions in this manuscript. The considerations were accepted and certainly raised the scientific level of this article. The modifications are highlighted in yellow stripes on the text.
Author: The results of the retrieved studies were obtained from meat (beef, chicken, pork, poultry and turkeys), fish, shellfish and broth media samples (lines 28-29).
Cold storage effects on Salmonella behavior (log CFU reduction or growth) in meat (beef, chicken, pork, poultry and turkeys), fish, shellfish and in broth media were the specified population. (lines 103-104).
Dear Editor, Please see the attachment, the OSF platform recommendation
